



# Population dynamics and reproduction strategies of planktonic foraminifera in the open ocean

Julie Meilland[1*], Michael Siccha[1], Maike Kaffenberger[1], Jelle Bijma[2], Michal Kucera[1]

[*] corresponding author: jmeilland@marum.de

[1] MARUM Center for Marine Environmental Sciences, University of Bremen, Leobener Straße 8, 28359 Bremen, Germany

[2] Alfred Wegener Institute Helmholtz Centre for Polar and Marine Research, Bremerhaven, Germany

**Abstract**

It has long been assumed that the population dynamics of planktonic foraminifera is characterised by synchronous reproduction associated with ontogenetic vertical migration. However, due to contradictory observations, this concept became controversial and subsequent studies provided evidence both in favor and against these

phenomena. Here we present new observations from replicated vertically resolved profiles of abundance and shell size variation in four species of planktonic foraminifera from the tropical Atlantic to test for the presence, pattern and extent of synchronised reproduction and ontogenetic vertical migration in this oceanic region. Specimens of *Globigerinita glutinata*, *Globigerinoides ruber ruber*, *Globorotalia menardii* and *Orbulina universa* were collected over the first 700 m resolved at nine depth intervals at nine stations over a period of 14 days. Dead

specimens were systematically observed irrespective of the depth interval, sampling day and size. Conversely, specimens in the smaller size fractions dominated the sampled populations at all times and were recorded at all depths indicating that reproduction might have occurred continuously and throughout the occupied part of the water column. However, a closer look at the vertical and temporal size distribution of specimens within each species revealed an overrepresentation of large specimens in depths at the beginning of the sampling (shortly after

the full moon) and an overrepresentation of small individuals in surface and subsurface by the end of the sampling (around new moon). These observations imply that a disproportionately large portion of the population followed for each species a canonical reproductive trajectory, which involved synchronised reproduction and ontogenetic vertical migration with the descent of progressively maturing individuals. This concept is consistent with the initial observations from the Red Sea, on which the reproductive dynamics of planktonic foraminifera has been modelled.

Our data extend this model to non-spinose and microperforate symbiont-bearing species, but contrary to the extension of the initial observations on other species of foraminifera, we cannot provide evidence for ontogenetic vertical migration with ascent during maturation. We also show that more than half of the population does not follow the canonical trajectory, which helps to reconcile the existing contrasting observations. Our results imply that the flux of empty shells of planktonic foraminifera in the open ocean should be pulsed, with disproportionately

large amounts of disproportionately large specimens being delivered in pulses caused by synchronised reproduction. The presence of a large population reproducing outside of the canonical trajectory implies that individual foraminifera in a fossil sample will record in the calcite of their shells a range of habitat trajectories, with the canonical trajectory emerging statistically from a substantial background range.


## 1. Introduction

The concept of synchronous reproduction followed by a predictable adjustment of depth habitat during ontogeny (ontogenetic vertical migration) has been the paradigm of population dynamics in planktonic foraminifera for more than half a century. Guided by instructive cartoons in publications and textbooks (Figure 1 and reference therein), researchers have subsequently applied these concepts to interpret short-term variability in flux of empty shells (e.g., Lin, 2014; Jonkers et al., 2015; Venancio et al., 2016) and to estimate the position in the water column where the final chambers of the shells with their wealth of geochemical proxies have been produced (e.g., Steinhardt et al., 2015; Takagi et al., 2016). The paradigm is tightly linked with the notion that planktonic foraminifera are obligate sexual outbreeders (Hemleben et al., 1989). Indeed, this unusual reproductive strategy among unicellular plankton, is congruent with, and perhaps even reliant on, temporally and spatially coordinated release of gametes (Weinkauf et al., 2020).

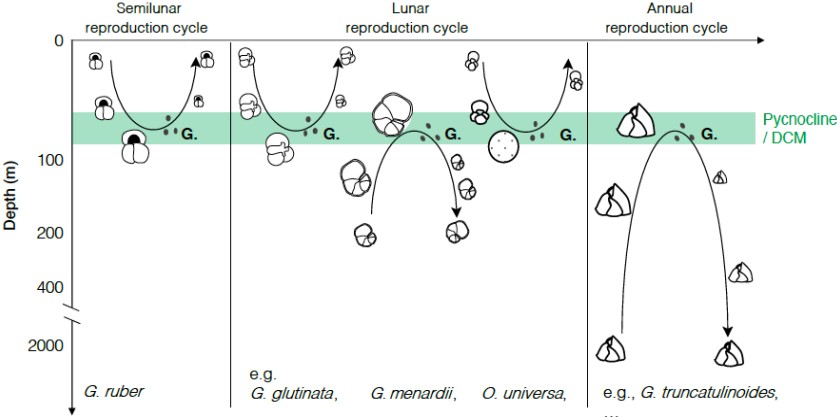

Figure 1: Canonical view of the reproductive strategy of selected species of planktonic foraminifera with synchronised reproduction and ontogenetic vertical migration, modified after the idealised schemes proposed by (Hemleben et al., 1989; Schiebel and Hemleben, 2005, 2017) ."G." indicates gametogenesis with black dots depicting the gametes.

Synchronised reproduction is known to occur among many marine organisms, such as corals, which are observed to spawn once a year synchronously within and even between communities, sometimes located hundreds of kilometers away (Babcock et al., 1986). Synchronous gamete release has also been observed in pelagic organisms such as marine algae (Brawley and Johnson, 1992). This type of synchronisation requires either an efficient internal biological clock, or it can be, less efficiently but more economically, triggered by external cues, such as the annual seasonal cycle or by periodic changes in nighttime illumination and tides, both linked to the lunar cycle (Clifton, 1997; Žuljević and Antolić, 2000).

Rhumbler (1911) was among the first to observe mass release of gametes in planktonic foraminifera and this phenomenon has been since confirmed by laboratory observations in many species (Anderson and Bé, 1976 and reference therein; Be et al., 1977; Hemleben et al., 1989and reference therein). That the mass release of gametes (hundreds of thousands, Spindler et al., 1978) may be synchronised was first noticed for the species *Hastigerina*



*pelagica* (Spindler et al., 1979). Here, in-situ observations (Almogi-Labin, 1984) and laboratory experiments (Spindler et al., 1979) showed a strong periodicity apparently aligned with the synodic lunar cycle, but driven internally as it was observed in specimens kept in laboratory, without exposure to any obvious lunar-cycle related cues. Further research provided evidence for synchronised reproduction in other species of planktonic foraminifera, based on observations in the Red Sea (*Trilobatus sacculifer, Globigerina siphonifera and Globigerinoides ruber*; Bijma et al., 1990; Erez et al., 1991; Bijma and Hemleben, 1994; Hemleben and Bijma, 1994) and in the North Atlantic (*Globigerina bulloides, Neogloboquadrina pachyderma* and *Turborotalita quinqueloba;* Schiebel et al., 1997; Volkmann, 2000; Stangeew, 2001). Observations of a periodicity (lunar, semilunar or even annual, Figure 1) in foraminifera fluxes from sediment trap samples for e.g., the species *Orbulina universa, Globigerinella siphonifera* (*aequilateralis*) and *Globorotalia menardii* (Kawahata et al., 2002; Jonkers et al., 2015) further corroborated the notion that the reproduction of many species of planktonic foraminifera in the upper ocean is synchronised by periodic cues, related to the lunar cycle.

Next to the observation of synchronised reproduction, analyses of vertically resolved plankton tows from the Gulf of Eilat and central Red Sea led (Erez et al., 1991) and (Bijma and Hemleben, 1994) to introduce the concept of concerted vertical shift in the habitat of the synchronously reproducing population: the ontogenetic vertical migration (OVM). The notion that the reproductive cohort undergoes a concerted vertical movement (typically sinking) terminated by reproduction at a specific depth would further increase the chance of gamete fusion, by co-locating their synchronous release in space. The mechanism facilitating such an orchestrated vertical migration is at hand: as the individual foraminifera grow, the weight of the calcite shell increases disproportionately, generating sufficient negative buoyancy to counteract passive movement by turbulence in the mixed layer and the increasingly adult specimens embark on a collective descent to the reproduction depth (Erez et al., 1991). Like the concept of synchronised reproduction, OVM appeared to be supported by observations of distinct geochemical signatures associated with the final chambers of foraminifera shells, indicating that these were produced in different (typically deeper) parts of the water column than the rest of the shell (e.g., Pracht et al., 2019).

As a result, the (lunar, semilunar or annual) synchronous reproduction model associated with OVM has been generalised for most species of planktonic foraminifera (Figure 1). These appealing and logical schemes became widely adopted, so much that we almost omitted that the presented depictions of life cycles of most species were idealised, still required additional observations and that there remains a host of problems and uncertainties, challenging the presented models. Next to the discovery of active photosymbiosis (Takagi et al., 2020) in species whose hypothetical OVM trajectories extend below the photic zone, there have been numerous observations of stable vertical habitats in the plankton (Rebotim et al., 2017; Iwasaki et al., 2017; Greco et al., 2019; Lessa et al., 2020) as well as shell flux patterns in sediment traps (Lončarić et al., 2005; Chernihovsky et al., 2020) showing no evidence of OVM and reproduction synchronised by the lunar cycle. Even where plankton tow and sediment trap could be interpreted as indicative for cohort growth (synchronised reproduction), the timing of reproduction with respect to the lunar cycle appeared to vary within species and among species (e.g., dephasing between *G.menardii* and *G. siphonifera* in Jonkers et al. (2015); reproductive event suspected after the full moon in Venancio et al. (2016); continuous growth of *T. sacculifer* up to 7 days after full moon in Jentzen et al. (2019)) and the external cue responsible for the synchronisation remained unclear.

On a more conceptual level, the OVM phenomenon suffered from the lack of explanation for habitat depth restitution after gametogenesis. Whereas it can be easily shown that the adult, often "sinking", part of the



postulated OVM cycle is mechanically plausible, how the tiny (10-20 μm) gametes and zygotes succeed to ascend the same distance as the descending adults over a period of days, to emerge as juvenile specimens intercepted by

plankton nets at the surface remained unanswered. Similarly, the apparent necessity to synchronise gamete release in space and time to facilitate reproduction has been challenged by reports of the existence of an asexual mode of reproduction in at least two species of planktonic foraminifera from different clades (Takagi et al., 2020; Davis et al., 2020). Although the canonical concept of reproduction dynamics in planktonic foraminifera has been derived from observations, it remains unclear whether it applies to all species and whether it takes place at all times. This

uncertainty is largely due to the lack of direct observational data obtained in the open-ocean habitat of planktonic foraminifera, that would reproduce the initial observations from the Red Sea and allow a direct assessment of reproduction dynamics for more species.

This is unfortunate considering the consequences (lunar) synchronised reproduction and OVM have on the calibration of models simulating planktonic foraminifera population growth, for biogeochemical cycles and

paleoproxies. With their calcite shell, planktonic foraminifera are major carbonate producers in the pelagic environment (e.g., Schiebel, 2002). Synchronised reproduction would generate pulses in the export flux of larger shells to the deep ocean and therefore impact marine biogeochemical cycles (Kawahata et al., 2002). Since foraminifera shells grow by sequential addition of chambers, the shell incorporates a sequence of chemical characteristics of all environments where the growth took place. Therefore, interpretations of the geochemistry of

the shells require knowledge of where specimens grew. A widespread and extensive OVM in planktonic foraminifera populations, whether it concerns the entire population or just a small percentage of it, would generate inhomogeneity in the chemical composition of the concerned shells, which may complicate the interpretation of the overall signal and especially of the signal recorded by individual shells.

The reproductive strategy of planktonic foraminifera can be best addressed by direct observations, capturing both

the temporal and the vertical (ontogenetic migration) dimension. Such observations require vertically resolved sampling of the same population, replicated in time over the entire period of the proposed reproduction periodicity. Both requirements are hard to achieve, with oceanographic expeditions typically covering linear transects, rather than remaining for weeks within the same water mass. Serendipitously, we were able to obtain a set of samples suitable to address at least to some degree the reproduction strategy of planktonic foraminifera in the North East

subtropical Atlantic ocean during the M140 cruise (Kučera et al., 2019). For a period of two weeks, the ship remained in a similar region and we were able to obtain a set of vertically resolved plankton samples which allows us to test the existence of a temporally synchronised reproduction and the presence of OVM in planktonic foraminifera in the open ocean. To this end, we measured and analysed the abundances and size distribution of four species *Globigerinita glutinata*, *Globigerinoides ruber ruber*, *Globorotalia menardii* and *Orbulina universa*,

representing all three main clades of extant planktonic foraminifera.

## 2.  Material and methods

### 2.1.  Sampling

Because of the necessity to service two of the eastern sediment traps and moorings of the NIOZ trans-Atlantic

array (Stuut et al., 2019), the R/V METEOR remained during the first two weeks of the cruise M140 (Kučera et al., 2019) within similar water masses in the eastern part of the tropical Atlantic ocean, north off 10°N (Figure 2). This part of the Cape Verde Basin is characterised by the presence of the North Equatorial Current (NEC), the



Guinea Dome (GD) in the South East and the Intertropical Convergence Zone (ITCZ) in the South (Figure 2). During the two weeks of the cruise, the population of planktonic foraminifera in the water column was sampled

at nine stations (Figure 2). Every day at 8:00 (local time) from the 12[th] to 14[th], the 17[th] to 21[st] and on the 25[th] of August 2017, two successive sampling casts were carried out, using a multi-plankton-sampler (MPS, Hydrobios, Kiel) equipped with five nets (100 µm mesh size, 0.25 m² opening) closing sequentially at successive discrete depths during the upcast. The first and deeper cast collected water from the intervals of 700-500, 500-300, 300-200, 200-100, 100-00 while the second cast resolved the upper layer at intervals of 100-80, 80-60, 60-40, 40-20,

20-0 (Table 1). The resulting 81 samples (9 vertical profiles resolved at 9 depth levels) were processed either directly on board or stored by freezing and processed later. The samples were analysed without splitting. All specimens of planktonic foraminifera were picked and identified to the species level following the taxonomy of Schiebel and Hemleben (2017). Individuals bearing cytoplasm were assumed living and counted separately from what was considered as empty shells. Following Morard et al. (2019) we use *Globigerinoides ruber ruber* instead

of the commonly used *Globigerinoides ruber* pink.

Table 1: Station number (in brackets their reference name in Kučera et al. (2019)), location (degree decimals), date of sampling and corresponding day in the lunar cycle (LD). The studied sampling depth intervals for all stations are 0-20, 20-40, 40-60, 60-80, 80-100, 100-200, 200-300, 300-500 and 500-700 m depth.

| Station | Latitude | Longitude | Date | Lunar day |
|---|---|---|---|---|
| 2 (GeoB22402) | 15.871 | -28.745 | 12/08/17 | 20 |
| 3 (GeoB22403) | 14.791 | -32.507 | 13/08/17 | 21 |
| 4 (GeoB22404) | 13.721 | -36.221 | 14/08/17 | 23 |
| 6 (GeoB22406) | 12.293 | -36.942 | 17/08/17 | 26 |
| 7 (GeoB22407) | 12.096 | -36.794 | 18/08/17 | 27 |
| 8 (GeoB22408) | 12.099 | -30.378 | 19/08/17 | 28 |
| 9 (GeoB22409) | 11.880 | -26.662 | 20/08/17 | 29 |
| 10 (GeoB22410) | 11.437 | -22.818 | 21/08/17 | 0 |
| 12 (GeoB22412) | 14.112 | -23.740 | 25/08/17 | 4 |


The MPS was equipped with a pressure sensor, allowing net opening at precisely determined depths, a flow meter to determine directly the volume of water filtered by each net, later used to estimate planktonic foraminifera concentrations and a Conductivity Temperature Depth (CTD) probe (SST CTDM90), recording CTD profiles at each station except for station 7 for which no profile was recorded. The CTD profiles showed overall similar

physical conditions in the water, albeit with lower subsurface temperatures and salinities in the Eastern part of the region and a shallower mixed layer depth (MLD), above 50 m depth (Figure 2). The chlorophyll-*a* concentrations remained low (< 1.5 mg.m⁻³) and the deep chlorophyll-*a* maximum (DCM) was located below the MLD (deeper DCM found in station 4 and shallower in station 10 with respective depth of 83 and 45 m, Figure 2).




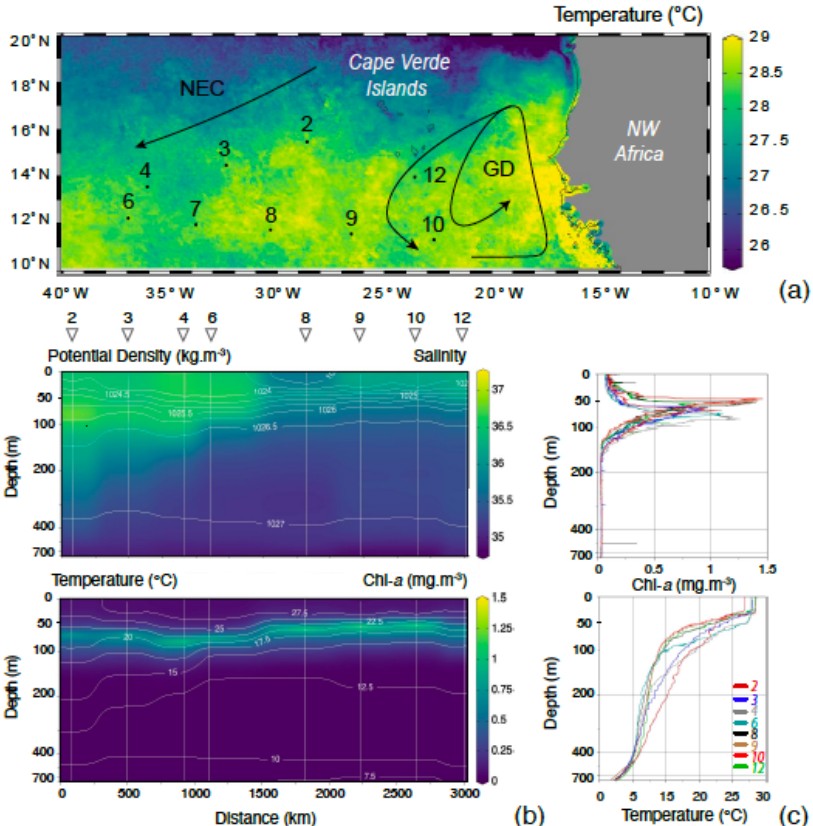

Figure 2: a.) Map of the study area showing the position of the studied vertical profiles (black dots with numbers), remotely sensed sea-surface temperature (Z-axis; August 2017, aqua-MODIS 4 km resolution, data produced with the Giovanni online data system, developed and maintained by the NASA GES DISC), the North Equatorial Current (NEC) and the Guinea Dome (GD) highlighted by black arrows (after Fieux (2021) and reference therein). b.) CTD Salinity section with isopycnals (white, ranging from 1023 in surface to 1027 below 300 m) and Chl-*a* concentration (mg.m⁻³) with isothermals (white, ranging from 27.5 in surface to 7.5°C below 600 m) and with stations highlighted by straight vertical white lines, triangles and numbers on top of the plot (no CTD profile was recorded for station 7). c.) Station profiles (colored lines) of Chl-*a* concentration (mg.m⁻³) and Temperature (°C). For all plots, the x-axis is stretched in the top to allow a better observation of the Mixed Layer Depth (MLD). Figures were drawn by Ocean Data View (Schlitzer, 2015).

### 2.2. Planktonic foraminifera size measurements

From among the identified and counted species, the three most consistently occurring among the stations were used to test for the existence of synchronised reproduction and OVM by measuring their shell sizes. All specimens (cytoplasm-bearing and empty) of *Globigerinoides ruber ruber* (n=1073), *Globorotalia menardii* (n=1085) and *Globigerinita glutinata* (n=3520) were manually transferred to customized microslides (Kreativika), oriented in the umbilical view and imaged using a Keyence digital microscope (VHX 6000) equipped with 100-1000x



magnification objective (VH-Z100R), an automated stage (VHX S650E) and an LED ring light (OP-88164). Size
measurement calibration was performed automatically by the microscope by use of a calibration stage inset.
Imaging was performed in confocal depth composition mode and magnifications ranging between 200 and 500x.
The acquired images and elevation data were analysed with a custom script (MATLAB 2017b). The automated
segmentation of foraminifera from the background occurred in two steps. A coarse primary segmentation was
obtained by using only the elevation data to isolate particles from the background based on their height. The final
particle segmentation was generated using an implementation of the sparse field method by Lankton (2009) on
the RGB data with the previously generated segmentation mask as the seed area. Particle measurements were then
performed on the final segmentation mask using the MATLAB Image Processing Toolbox. Among the available
size parameters we systematically used the minimum diameter for the analyses (further developed in 2.3). This
allows us to identify specimens that are effectively of a similar size or larger than the size of the net mesh (100
µm). This is important because foraminifera smaller than the nominal mesh size often remain in the catch, because
of clogging of the net or because the small specimens adhere to larger particles. Finally, the segmentation masks
were manually checked for the presence of broken specimens and overlapping particles (foraminifera touching
each other), which were all then removed from the dataset. In addition to the three species, we also analysed the
occurrence of juvenile (n=62) and terminal (n=250) ontogenetic stages of the species *Orbulina universa*, which
can be easily separated by the presence of the terminal spherical chamber.

### 2.3.    Data analysis

To compensate for the low abundances of specimens in the deeper net intervals and in order to treat the data in a
coherent way with regard to the environmental parameters (e.g. depth of the MLD, see 2.1), we performed
similarity analysis and grouped depth level intervals using "absolute data" and "complete linkage". This led to a
reduction of the original nine depth intervals into five new intervals: surface 0-20 m, supra epipelagic 20-40 m,
epipelagic 40-80 m, infra epipelagic 80-300 m and mesopelagic 300-700 m.

Because of the underlying (and expected) lognormal distribution of size and the fact that the sampling begins at
100 microns (size of the mesh), the size distributions are artificially truncated and rather than trying to compensate
for this by transformations, we decided to rely on non-parametric approaches in the analyses of such data. We
therefore employed the Kruskal-Wallis test instead of ANOVA to assess the potential variability in abundance
and size of the studied species across the five depth intervals and along the nine vertical profiles. Additionally,
overall differences in the distribution of the data without any transformations were tested by a discrete
Kolmogorov-Smirnoff test. The statistical tests were formulated to evaluate specific hypotheses for *Globigerinita
glutinata*, *Globigerinoides ruber ruber* and *Globorotalia menardii* only (specimens of *Orbulina universa* were
not specifically measured but separated in the two categories "juvenile" and "adult" and did not occur in
concentrations allowing statistical testing).

In the absence of synchronised reproduction, we would expect no significant differences in the shape of the overall
size distribution among stations and random variability in planktonic foraminifera concentrations among the
stations (for living and for dead specimens), reflecting patchiness (Siccha et al., 2012; Meilland et al., 2019).
In the absence of OVM within planktonic foraminifera species, we would expect for each species no significant
differences in the shape of the size distribution across the depth intervals covering the habitat depth of the species.



In addition to the above mentioned analyses, we also visualised the differences in the proportion of shells of a
certain size occurring at a certain depth or time following the concept of Bijma et al. (1990). To this end, we
transformed the analysed proportions across stations and sampling intervals into residuals of size classes,
highlighting size classes that are over- (positive residuals) and underrepresented (negative residuals) at certain
stations or sampling intervals:

$$\text{Residuals}_{\text{Station}} = (P_{\text{ST SCi}}) - (\text{mean} (\textstyle\sum_{i}^{N} P_{\text{ST SCi}}))$$

$$\text{Residuals}_{\text{Depth}} = (P_{\text{SP SCi}}) - (\text{mean} (\textstyle\sum_{i}^{N} P_{\text{SP SCi}}))$$

P = Percentage of foraminifera

SC = Size-class based on specimens minor-axis

ST = Sampling station

SP = Sampling interval

Following the method of Bijma and Hemleben (1994), the residuals were calculated on the sum of cytoplasm-
bearing and empty shells, but because the proportion of the latter was low within the living zone, the results would
be similar even if the analysis was limited to the cytoplasm-bearing shells.

In addition, we calculated "relative mortality" (%) per size-class following the method described in Hemleben
and Bijma (1994) as:

$$\text{Relative mortality in SC1} = ((RA_{\text{SC1}} - RA_{\text{SC2}})/RA_{\text{SC1}}) * 100$$

RA = mean relative abundance of foraminifera

SC = Size-class based on specimens minor-axis, with 1 being the smaller one to X being the larger.

In a scenario of synchronised reproduction, this calculation uses the difference of foraminifera abundances
between a smaller size fraction and a larger size fraction to estimate pre-adult mortality in the small size-fractions
and distinguish it from what could be attributed to post-reproduction mortality in the larger size fractions.
"Residuals" and "Relative mortality" were calculated for foraminifera with minimum diameter larger than 100
µm, to avoid potential bias due to differential net clogging. "Relative mortality" was not calculated for *Orulina*
*universa* as specimens were not measured but separated in only two categories: "Juveniles" and "Adults".

## 3. Results

### 3.1. Planktonic foraminifera abundances, distribution and size

Over the sampling area maximum concentrations of the three species of planktonic foraminifera *Globigerinita*
*glutinata*, *Globigerinoides ruber ruber* and *Globorotalia menardii* were observed between 0 and 40 m and at the
end of the cruise while specimens of *Orbulina universa* were most abundant above 60 m with highest abundance
at the beginning of the cruise. *G. glutinata*, *G. ruber ruber* and *G. menardii* have a clear surface maximum reaching
up to 79, 47 and 122 ind. m$^{-3}$ respectively between 0 and 20 m in station 12. The highest concentration of *O.*
*universa* (12 ind. m$^{-3}$) was observed at station 2, between 40 and 60 m (Figure 3a). As shown in Figure 3b, the
occurrence of foraminifera was spread over several sampling depth intervals for the first stations and narrowed
down to the two shallowest depth intervals by the end of the cruise, in station 10 and 12. This is particularly visible
for *G. menardii* and *O. universa*. The ratio between living and dead specimens (Figure 3c) shows a decreasing
proportion of living specimens with depth at all stations. For *G. glutinata*, the proportion of dead specimens is



particularly high below 60 m. For all other species this occurs deeper in the water column, below 100 m (Figure 3c).

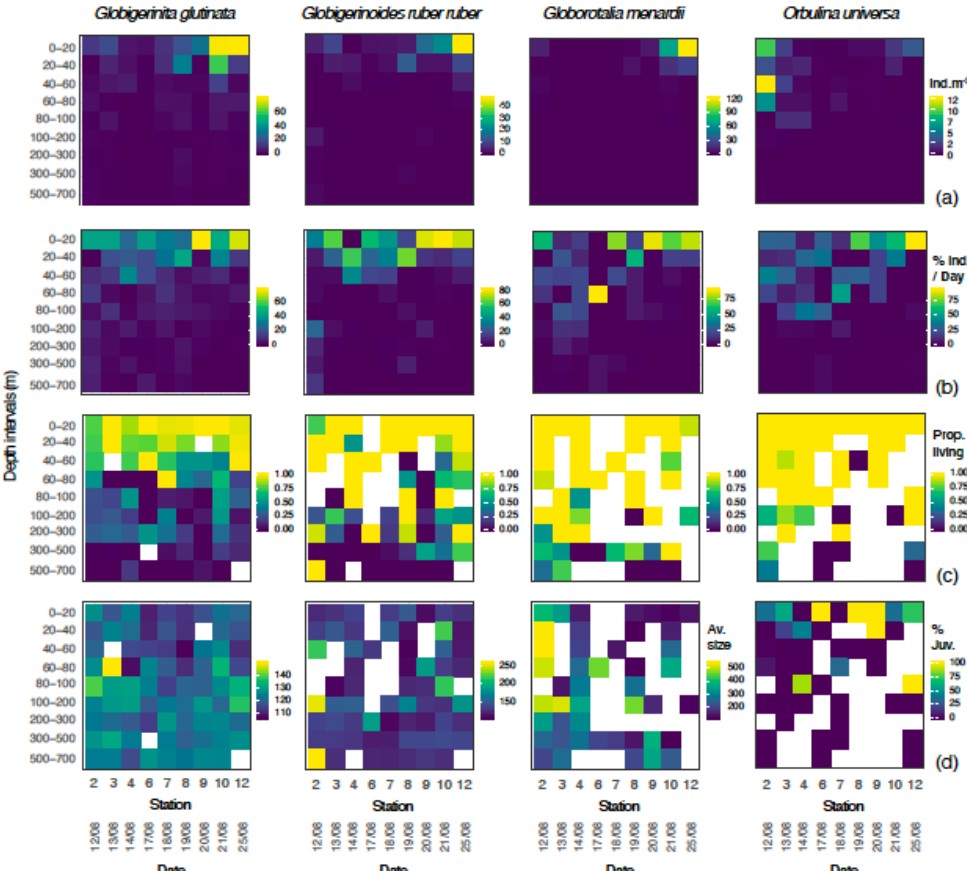

Figure 3: *Globigerinita glutinata*, *Globigerinoides ruber ruber*, *Globorotalia menardii* and *Orbulina universa* (a) abundances (ind. m⁻³), (b) normalized abundances per day (%), (c) ratio between living and dead specimens and (d) average size based on the minor axis (μm) and excluding all specimens smaller than 100 μm, per depth interval

(y-axis) and per station and date (x-axis). White squares represent the absence of specimens. For *O. universa* panel (d) displays the percentage of juveniles with yellow = juveniles only and dark blue color = adult specimens only.

Overall, specimens of *G. glutinata*, *G. ruber ruber* and *G. menardii* were rather small with respective median size of 114, 114.5 and 116 μm or 119, 122 and 135 μm when only considering specimens larger than 100 μm to avoid

a potential net clogging bias (Figure 3d; Table A1, A2 and A3). Specimens of *G. glutinata* appear to be slightly larger below 60 m for every station, especially for station 3 where the maximum average size is observed between 60 and 80 m (149.4 μm). The largest specimens of *G. ruber ruber* were observed at station 2, between 100 and 200 m and between 500 and 700 m and with respective average size of 253 and 259 μm but otherwise no clear tendency emerged. Larger individuals of *G. menardii* were recorded at station 2, from 20 to 200 m with an average

size larger than 500 μm (534 μm in 20-40 m, 539 μm in 40-60, 508 μm in 60-80 and 501 μm in 100-200 m depth),



strongly contrasting with the small size of specimens collected for every other station and particularly larger than the individuals collected at the end of the cruise (below 140 μm in station 12). For *O. universa*, a larger proportion of juveniles was observed between 0 and 40 m in stations 6, 8 and 9. In station 4 and 12 juveniles of *O. universa* were dominant between 80 and 100 m (Figure 3d).


### 3.2. Statistical analyses for synchronised reproduction

The statistical analysis for the abundance and size data of *Globigerinita glutinata*, *Globigerinoides ruber ruber* and especially of *Globorotalia menardii* (living and dead) do not provide evidence against the existence of synchronised reproduction among the studied populations of these species.

Based on a discrete Kolmogorov-Smirnoff test, the size distribution in station 10 and 12 statistically differ from the distributions observed at (almost) all other stations. Specifically, living specimens of *G. glutinata* were significantly larger at the beginning of the sampling period than at the end. For example, individuals in station 2 were larger than in station 6, 7, 8, 9 and 12 and individuals in station 3 and 4 larger than in stations 6, 8 and 9. Specimens collected in station 8 were smaller than in stations 10 and 12 (Figure 4a, Table A4).

Significantly higher abundances of living specimens of *G. menardii* were observed in station 10 between 0 and 60 m. The specimens (dead and living) of *G. menardii* collected in stations 2, 3 and 4 were larger than those collected by the end of the cruise, for example in stations 8, 9, 10 and 12 (Table A4 and A5). Living individuals of *G. menardii* in station 12 were significantly smaller than those collected from station 2 to 4 but larger than those collected between station 8 to 10 (Figure 4a, Table A4). In station 10, living specimens of *G.menardii* were

significantly smaller than those collected at the beginning of the cruise from station 2 to 4 and in the last sampled station (12). We can therefore conclude that a significant "influx" of small living specimens of *G. menardii* occured in station 10.

Over the sampling area living individuals of *G. ruber ruber* were significantly more abundant in station 3, 9 and 10 (0 to 20 m), in station 8 (20 to 40 m) and in station 12 (0 to 40 m) than in the other stations and depth intervals.

Living specimens of *G. ruber ruber* were significantly larger in station 2 and 3 than in any other stations collected by the end of the cruise (from 7 to 12) where no significant differences in size could be observed (Table A4). The larger dead specimens of *G. ruber ruber* were also observed in station 2. These results indicate the presence of larger specimens of this species (living and dead) at the beginning of the sampling and higher abundances in surface waters from station 9 onwards.


### 3.3. Statistical analyses for ontogenetic vertical migration

For *G. glutinata* the mean size of living specimens between 0 and 20 m depth was significantly larger than between 40 and 80 m and significantly smaller than between 80 and 300 m (Figure 4b, Table A5). The mean size between 40 and 80 m was significantly smaller than in all other depth groups and the mean size between 80 and 300 m was

larger than in all shallower depth intervals. The size data for *G. glutinata* therefore do not show any clear signal of OVM. In contrast, we observe a successively increasing size for the living specimens with depth in *G. ruber ruber*. The size of the living individuals of *G. ruber ruber* was significantly larger below 40 m, (Figure 4b, Table A5) supporting the existence of OVM. Similarly, we also observed a successively increasing size with depth for *G. menardii*. Specifically, specimens collected between 0 and 20 m were smaller than below 40 m. Specimens

between 40 and 80 m were significantly larger than those collected between 0 and 20 m and specimens between



80 and 700 m depth were significantly larger than those collected between 0 and 40 m depth (Figure 4b, Table A2 and A5).

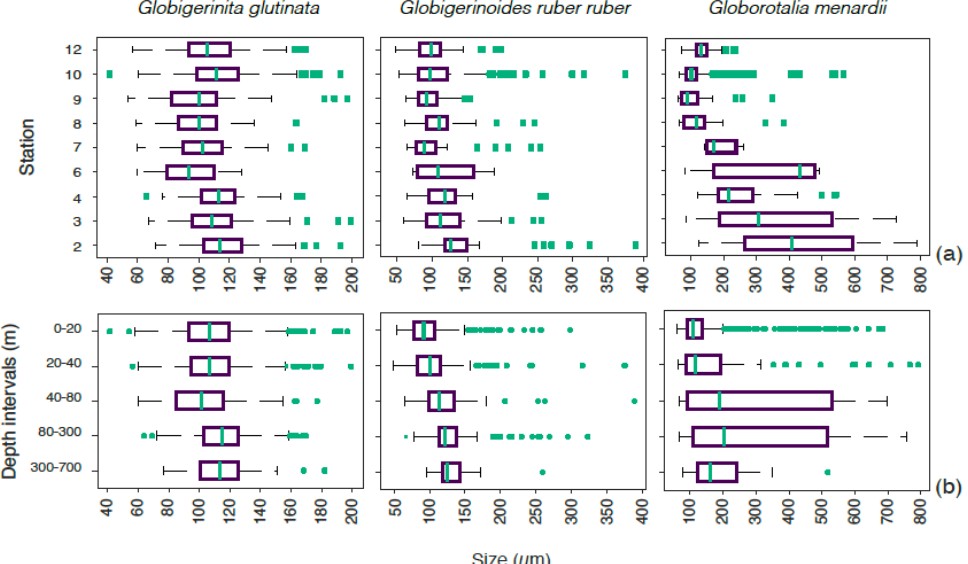

Figure 4: Size distribution of living specimens for the three studied species (a) across the different stations and (b) across the depth profile from the surface (0-20 m depth) to the mesopelagic environment (300-700 m depth). Green dots indicate outliers. The boxes extend to the interquartile range (IQR) and the whiskers indicate 1.5*IQR. The green lines in the boxes indicate the median.

### 3.4. Disproportional occurence of foraminifera within specific size fractions over time (synchronised reproduction) and depth (OVM)

Visualisation of the residuals of foraminifera occurrence among the different size fractions with time and depth allows for a quick identification of size classes that are over- and underrepresented at certain stations or depth intervals in comparison to the whole dataset (Figure 5). As explained in the material and method section (2 d) residuals were calculated for specimens larger than 100 µm only and therefore excluded about 22 % of the collected individuals of *G. glutinata*, 28 % of *G. ruber ruber* and 31 % of *G. menardii* (Table A1). For *Orbulina universa* we treated "Juvenile" and "Adult" specimens as two size fractions and calculated the residuals accordingly.

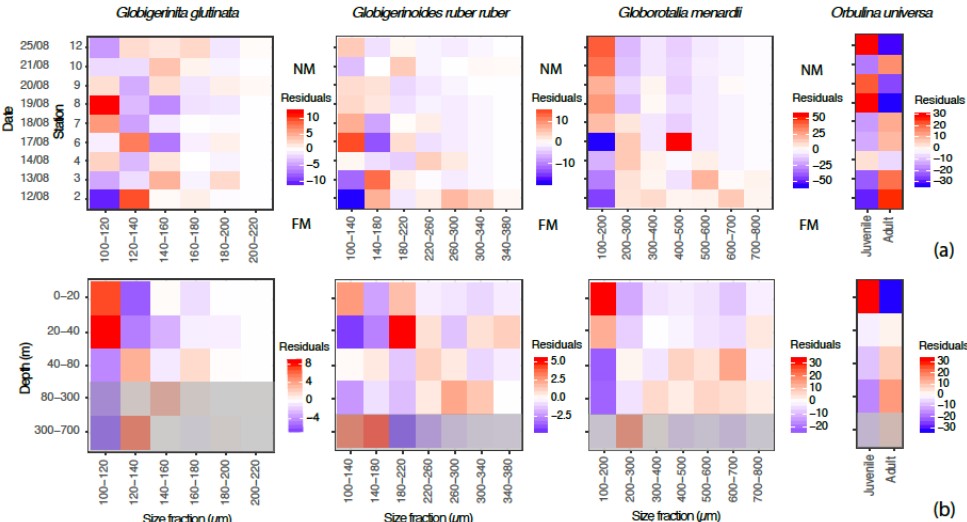

Figure 5: Proportions of specimens of the three studied species of planktonic foraminifera expressed as residuals,
highlighting size classes that are over-represented (red tones) or under-represented (blue tones) for specific (a)
days (y-axis) in comparison to the whole dataset with FM and NM highlighting the time of the Full Moon and of
the New Moon; (b) depth intervals (y-axis) in comparison to the whole dataset. The grey filter indicates the depths
where > 50% of the specimens were dead (empty shells) and the intercepted filled shells therefore likely represent
background mortality across the different size fractions (Peeters and Brummer, 2002).

For the specimens of *G. glutinata* small specimens appear disproportionately more abundant at the beginning of
the cruise at station 2 and then again at stations 7 and 8 (Figure 5a). Across the different depth intervals (Figure
5b) we can notice an overrepresentation of small individuals (<120 μm) and underrepresentation of larger
individuals above 40 m. Below 40 m, the opposite pattern emerges, with a systematic underrepresentation of
specimens in the smaller size fraction and an overrepresentation in the larger ones (120-200 μm).

Over time, large specimens of *G. ruber ruber* (> 220 μm) appear overrepresented at the beginning of the cruise
(station 2 and 4) and underrepresented after six days (from station 6 and 7). Conversely, very small specimens
(<140 μm) are overrepresented from the sixth day of sampling, from station 6 (only exception for station 10,
Figure 5a). Vertically, individuals of *G. ruber ruber* larger than 180 μm are overrepresented below 0 m depth
within the production zone (Figure 3c, 5b).

Residuals for the specimens of *G. menardii* show a clear signal over time and vertically. Over time, large
specimens (>200 μm) are overrepresented during the first six days of sampling (station 2 to 6) with very large
specimens (>500 μm) being overrepresented during the first two days (station 2 and 3, Figure 5a). From the
seventh day of sampling the trend is inverted with a clear overrepresentation of small specimens (<200 μm) until
the last day of the sampling. Vertically, these small specimens are only overrepresented between 0 and 40 m,
below which larger specimens (>200 μm) are systematically taking over.

Because the concentrations of *O. universa* along the study were lower than for the other species (Figure 3a), its
residuals should be treated with care. However, juveniles of *O. universa* are clearly overrepresented from station



8 to 12 and in surface water, from 0 to 20 m while "adults" are overrepresented during the first half of the cruise

and below 40 m depth.

Observation of the residuals of *G. glutinata*, *G. ruber ruber*, *G. menardii* and *O. universa* over the depth intervals (Figure 5b) therefore support the existence of an OVM, with larger specimens overrepresented below 20 m (*G. ruber ruber*) and 40 m (*G. glutinata*, *G. menardii* and *O. universa*) and smaller (living) specimens overrepresented in surface. A pattern of synchronicity in the reproduction cannot be excluded and residuals of *G. ruber ruber* and

especially of *G. menardii* and *O. universa* supports such a model with an overrepresentation of large specimens (potential mature adults) shortly after the full moon, at the beginning of the cruise, and of small specimens (potential juveniles) around the new moon, by the end of it.

### 3.5.    Mortality/population loss and expected size of maturity

Assuming our sampling intercepted a largely similar regional population, affected in abundance only by patchiness, with reproduction synchronised in the same way, the obtained size data could be used to estimate the size class where reproductive mortality begins (Figure 6; Hemleben and Bijma, 1994; Schiebel et al., 1997). To determine the size class in which one could expect maturity and higher chances of reproduction, we determined the loss rate of each size-class in the first 100 m (production zone) across all stations and generated histograms of

relative mortality for each species (Figure 6). For the three species over the studied area, especially for *G. menardii*, the population was largely dominated by the presence of foraminifera in the smaller size-class (Figure 6a), with abundance decreasing with size. Because of the overproduction of gametes per individual, the mortality in planktonic foraminifera is expected to be very high among the smallest size class (youngest individuals, Brummer and Kroon, 1988) and decrease until a point when maturity is reached and mortality increases until

100% in the largest size class. This is because foraminifera terminate their life after gamete release (Be et al., 1977).

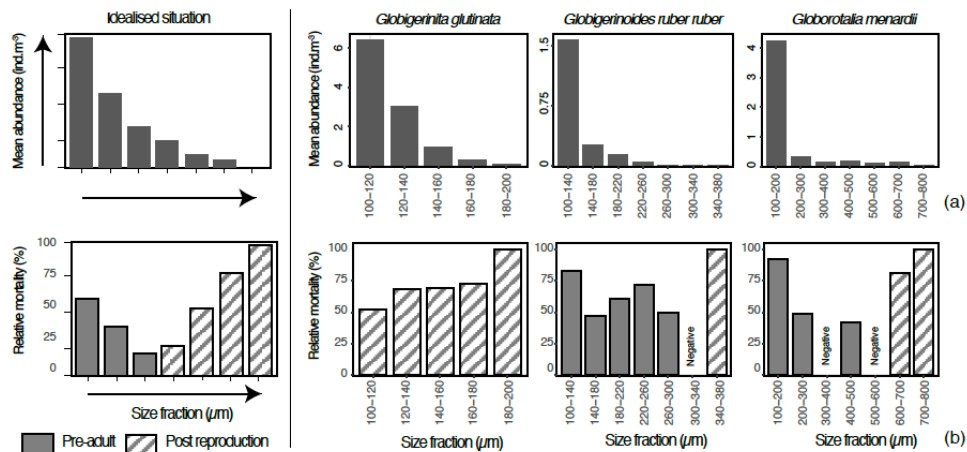

Figure 6 : Top and bottom-left plots show an idealised picture (model) of planktonic foraminifera mean abundance

and relative mortality per size fraction in the first 100 m depth with the size from which one can expect reproduction located at the junction of the "Pre-adult" and "Post reproduction" mortality (based on Bijma and




Hemleben, 1994; Schiebel et al., 1997). The three top and bottom-right plots show per species and size fraction (a) mean abundance (ind. m$^{-3}$) of planktonic foraminifera (all stations together) and (b) relative mortality (%) over the sampling time (see methods for the relative mortality calculation).


In all three analysed species, the apparent mortality profiles show patterns that can be explained in terms of the hypothetical models. For *G. glutinata*, mortality increases with size (Figure 6a), indicating that across all size classes we observe post reproduction mortality (i.e., mostly specimens who reproduced) and the minimum size of maturity is below the studies size range.

Mortality for *G. ruber ruber* is highest among specimens with a size between 100-140 µm (>77%) and then again among the specimens larger than 340 µm. Based on the hypothetical model, the high relative mortality in the smaller size fractions could be attributed to pre-adult mortality while the increase towards the larger size-fraction likely signals post reproduction mortality. The derived size from which reproduction of *G. ruber ruber* can be expected in our data would occur around 300 µm.

Higher relative mortalities for *G. menardii* are observed for specimens belonging to 100-200 µm and larger than 600 µm. Similarly to what is observed with *G. ruber ruber*, the relative mortality in the smaller size fractions could be attributed to pre-adult mortality while the ones in the larger size fractions likely highlight post-reproduction mortality. The inferred optimal size of reproduction for *G. menardii* would be around 500 µm.

**4. Discussion**

**4.1. Evidence for regionally synchronised reproduction**

The studied material comes from 9 stations covering a geographical area extending from 22.818°W to 36.942°W and from 11.437°N to 15.871°N (Figure 2a). Over the sampling area, the environmental parameters such as T°C and Salinity varied slightly (Figure 2c) and it is important to stress that our observations probably reflect the

dynamics of multiple populations of foraminifera rather than of a single one. Juveniles (foraminifera of small size fractions) were present throughout the entire survey in significant proportions for all species (Figure 3d and Figure 6a). Similarly, dead specimens of all species and all sizes were observed at every station (Figure 3c), never with a significantly higher or lower ratio. The constant presence of juveniles and dead specimens of foraminifera from all species suggest that reproduction may have occurred continuously during our survey.

However, at the beginning of the cruise (station 2 to 6) all four species occurred at low concentrations and their populations systematically comprised the largest specimens encountered throughout the entire cruise (Figure 3b and d, Figure 5a). Conversely, populations sampled by the end of the cruise (9 to 12) were characterized by higher concentrations of foraminifera (factor 10) and comprised smaller specimens (over-representation of the small size-fractions, Figure 5a). This is particularly visible in the residuals of *Globorotalia menardii* (Figure 6a) showing a

clear overrepresentation of large specimens (500-600 and 600-700 µm) in the two first sampled stations, visited four days after the full moon, and a clear overrepresentation of small specimens (100-200 µm) from station 7, (three days before the new moon). Similarly, the residuals of *G. ruber ruber* also indicate an overrepresentation of larger specimens on the first day of sampling and a dominance of small specimens from station 6. Indeed, the arrangement of the residuals for *G. ruber ruber* resembles the pattern that has been observed about 30 years ago

in the Red Sea and used to introduce the concept of synchronised reproduction (Bijma et al., 1990). Although not as clear, a signal of synchronicity in the reproduction can also be extracted from the size data of *G. glutinata*, with





the larger recorded specimens emerging the second day of sampling (Figure 3d) and an overrepresentation of the smaller size fraction in station 7 and 8 (Figure 6a). Finally, higher concentrations and an overrepresentation of mature adult specimens of *O. universa* (spherical chamber = imminent gametogenesis Caron et al., 1987) were

observed during the first days of sampling and from station 8, juveniles were dominant (Figure 5). All of these observations not only hint at the existence of synchronous reproduction in phase with the lunar cycle (Bijma et al., 1990).

Thus, despite persistent occurrence of cytoplasm-bearing juveniles throughout the monitoring period, the observed abundance and size patterns in the studied species are consistent with a significant portion of their populations

being involved in a synchronised reproduction. Although the sampling interval in this study is too short to robustly estimate the involved generation time and assess any potential periodicity, we note that in all species disproportionately large amounts of large specimens were observed shortly after the full moon and disproportionately large amounts of small ones were detected about 10 days later (Figure 5a) This observation is consistent with synchronisation by full moon, as initially suggested by Bijma et al. (1990) and observed in some

sediment trap time series (Jonkers et al., 2015). Since our sampling comprised populations covering a geographically large area, and we still observed hints for synchronised reproduction, it is likely that the synchronisation acted across a large part of the ocean, which would require triggering by an external cue or internal biological clock. In such a scenario, the synchronisation or reproduction in planktonic foraminifera would involve manypopulations, and be recorded consistently across large geographical areas. This is supported by the

observation that the calculated mortality profiles can be reconciled with the hypothetical model, which requires synchronous reproduction across the studied region. Such a large-scale synchronisation is also consistent with observations of synchronised reproduction in *G. bulloides* based on material collected over different years, seasons and geographical areas (Schiebel et al., 1997).

### 4.2. Evidence for ontogenetic vertical migration (OVM)

The concept of ontogenetic vertical migration suggests that adult specimens of planktonic foraminifera gather at a specific depth where they encounter optimum conditions, such as the pycnocline or the nutrient-rich deep chlorophyll-*a* maximum (DCM) (Murray, 1991), to release their gametes. In the OVM schemes presented in the literature (Figure 1), the adults either progressively sink during growth and the gametes released at depth ascend

to restitute the depth range of juvenile specimens (e.g. *G. ruber* and *O. universa*), or the adults ascend and the gametes released at the shallow reproduction depth descend (e.g. *G. menardii* and *G. truncatulinoides*).

In our data, specimens from the smallest size fraction dominated the populations at all times and all depths, which would intuitively speak against the existence of any kind of OVM (Figure 3d). However, the observed size distributions in all species can be reconciled with the existence of OVM, if we consider at what times and depth

certain size classes are overrepresented or underrepresented with regard to their overall mean abundance (Figure 5). Yet, our observations for *G. glutinata*, *G. ruber ruber*, *G. menardii* and *O. universa* only support OVM with sinking of maturing specimens followed by gamete ascent. This contrasts with the OVM pattern suggested for *G. menardii* by Schiebel and Hemleben (2017), according to which pre-adult specimens of this species inhabit a subsurface environment, ascending towards the end of their life for reproduction to the pycnocline/DCM, and the

released gametes descend to restitute the deeper habitat of the pre-adult population. Our data provide no support for this OVM pattern and our observations of the population remaining in the upper water layer, with descent of





adults to a depth of 80 m is consistent with the fact that *G. menardii* harbours symbiotic algae, which are photosynthetically active in specimens of all sizes (Takagi et al., 2019). Our samples included species representing all the major clades, so the pattern of OVM clearly does not appear to be limited to the spinose planktonic

foraminifera, from where all of the existing evidence to date originated. On the other hand, all of the analysed species bear symbionts, which means the observed reproductive strategy, or some aspect of it, such as the nature of the synchronisation by cues related to light, which would be sensed by the symbionts, or the direction, with descending adults and ascending gametes, could be specific to symbiont-bearing taxa. Asymbiotic and deep-dwelling taxa may follow other reproductive strategies or differently configured ontogenetic trajectories. Similarly

to the scheme of Schiebel and Hemleben (2017), and consistently with the numerous observations of species-specific "typical" living depths, our data indicate that for each species the OVM reaches to a different depth. Remarkably, in all cases, although the OVM reaches below the mixed layer, the habitat of the studied species remains systematically well above the DCM (Figure 2).

Irrespective of its direction and the depth where it occurs, the existence of ontogenetic vertical migration could be

advantageous for planktonic foraminifera. It may bring mature specimens away from predators (Erez et al., 1991), increase chances for gametes encounter (Weinkauf et al., 2020), and separate the habitats of maturing specimens and juveniles, which differ in size by an order of magnitude and likely follow different trophic strategies (Brummer et al., 1986). Also, the migration could even be the trigger for synchronous reproduction because descend through the water column induces a change in light intensity, and a certain threshold light level could act

as a cue inducing reproduction. To engage in an OVM trajectory with descend of maturing specimens, our hypothesis, as suggested by (Erez et al., 1991), is that rather than actively migrating, the foraminifera passively sink as their shells grow and the addition of new chambers and shell thickening progressively increases the effective density of the individual (e.g., Bé and Hemleben, 1970; Bé et al., 1980; Erez, 2003). This theory corroborates recent observations of increasing shell density with size and with depth for specimens of *Globigerina*

*bulloides* in the North Pacific Ocean (Iwasaki et al., 2019) and for living specimens of *Neogloboquadrina pachyderma* in the Barents Sea (Ofstad et al., 2021). The observed species-specific depth habitats (e.g., Bé, 1962; Fairbanks et al., 1982; Rebotim et al., 2017) would in this model emerge from differences in buoyancy due to different shell architecture and calcification, allowing the maturing specimens of different species to reach different target depth, where they remain until reproduction, with the resulting pattern being consistent with the

often observed species-typical habitat depths. In this model, most of the motion would happen within small size fractions of the population, escaping observation when the populations are sampled by coarse mesh sizes (>100 µm).

Next to the regulation of buoyancy by shell architecture and calcification, the descent of maturing individuals could be further achieved by 1) changes in the composition of the cytoplasm (such as changes in low-density lipid

concentration; Spindler et al., 1978), 2) changes in Reynolds number due to adjustment of the effective specimen size by rearranging the rhizopodial network (Takahashi and Allan, 1984), 3) the properties of fibrillar bodies hypothesised to help foraminifera maintain their vertical position (Anderson and Bé, 1976; Hemleben et al., 1989) or 4) through changes in the production of low-density osmolytes, as observed in marine phytoplankton (Boyd and Gradmann, 2002). Whereas there exist multiple candidate mechanisms to explain how larger planktonic

foraminifera can be found deeper and sustain themselves at specific depth intervals until reproduction, it is less easy to understand how gametes and/or juveniles can then rapidly ascend to restitute the shallowest habitat of the

OVM trajectory. Our observations indicate that the depth restitution would occur within 5 to 8 days of gamete release and involves vertical movement of 20-80 m. A large part of the movement would occur within the mixed layer (~40 m, Figure 1), where turbulence could facilitate dispersion of the very young and light juveniles, transporting sufficient numbers of them towards a shallow optimum depth, increasing their chances to survive and embark on the canonical OVM trajectory. However, for *G. glutinata* and especially *G. menardii*, the ascend begins below the mixed layer, requiring 20-40 m of active movement. This can be aided by positive buoyancy, facilitated e.g. by low-density osmolytes (Boyd and Gradmann, 2002), but the density differential for ascent is smaller than for descent, where the mineral ballast in combination with large size allows faster movement. Instead, we hypothesize that the ascent can be aided by the motility of the flagellated gametes. Sexual reproduction in planktonic foraminifera involves the release of thousands of flagellated, motile gametes that also contain energy reserves (Anderson and Bé, 1976; Be et al., 1977; Hemleben et al., 1989). These gametes are known to survive about a day (Hemleben et al., 1989) and can move with an estimated velocity of 25 - 100 $\mu$m. s$^{-1}$ (gamete speed estimations from Weinkauf et al., 2020) and therefore could theoretically traverse the required 10-40 m to reach the mixed layer in 24 hours, if moving perfectly vertically, e.g. due to phototaxis.

### 4.3. The canonical reproductive behaviour emerging among other reproductive dynamics

Although our data provide evidence for the existence of OVM and cannot exclude synchronous reproduction, in all taxa the signal of the typical "canonical" reproductive trajectory exists alongside a substantial noise, with dead and juvenile specimens occurring at all times throughout the water column (Figure 3c, Figure 6b). To estimate the percentage of the population that does not follow the canonical path of synchronised reproduction and OVM, we calculated the proportion of individuals found in all squares of the diagram in Figure 7 which are located outside of the canonical trajectory. The results suggest that about 75% of *O. universa* may follow the canonical trajectory of synchronised reproduction and the OVM, but based on much larger sample sizes and therefore offering more reliable estimates, more than 50% of *G. menardii* and up to 70% of *G. glutinata* and *G. ruber ruber* do not appear to take part in synchronised reproduction and about 50% of *G. glutinata* and more than 60% of *G. menardii* and *G. ruber ruber* do not appear to migrate vertically during ontogeny in the same way as the canonical trajectory.

Clearly, it is possible that the canonical reproductive behaviour emerges as a result of a stochastic pattern facilitated by the enormous overproduction of gametes. It has been previously estimated that only 5% of the offspring may reach a mature reproductive stage (Brummer and Kroon, 1988), likely because the majority of this 5% followed the optimal ontogenetic trajectory, whereas the rest are by chance stranded outside of the optimal trajectory and are less likely to reach maturity. This is supported in our data by high estimated mortality (>75%) in the smaller size fraction of *Globigerinoides ruber ruber* (<140 $\mu$m) and *Globorotalia menardii* (<200 $\mu$m, Figure 6b) that we attribute to selection among juvenile individuals dispersed along numerous ontogenetic trajectories, more or less suitable for survival and growth. As a result, only a small fraction of the pre-adult small specimens appear to reach the size range where reproductive mortality is observed (Figure 6a). This very large mortality was not observed in the smaller size-fraction of *Globigerinita glutinata* probably because the juvenile specimens of this small species were smaller than 100 $\mu$m, and remained below the detection limit of our sampling. Planktonic foraminifera whose trajectory follows the optimum ontogenetic trajectory for each species given its physiology (e.g. light for symbionts) are more likely to persist and a release of gametes at the optimum target depth where most of the successful specimens gather will more likely lead to the production of juveniles. Thus,



the synchronisation causes positive feedback, ensuring enough individuals will follow the optimum trajectory, but this is achieved on the cost of many individuals departing from it and making the canonical pattern hardly perceivable in observational data.

This partly explains why, as mentioned in the introduction, the model of synchronised reproduction has been challenged by the absence of recordable signal in many sediment trap and plankton tow studies (Lončarić et al., 2005; Rebotim et al., 2017; Iwasaki et al., 2017; Greco et al., 2019; Lessa et al., 2020; Chernihovsky et al., 2020). However, the concept of the emergence of the canonical trajectory from a large background juvenile mortality was raised already in the first study introducing the synchronisation concept in planktonic foraminifera (Bijma et

al., 1990). This aspect of the reproductive strategy model became progressively lost behind simplified schemes (Figure 1), which were constructed as instructive diagrams, but often interpreted literally, assuming that all specimens follow the depicted ontogenetic trajectory.

Finally, it remains unclear what role does asexual reproduction play in the overall population dynamics of the studied species. Recent observations for *Neogloboquadrina pachyderma* (Davis et al., 2020) and *Globigerinita*

*uvula* (Takagi et al., 2020) show that planktonic foraminifera can reproduce asexually. If the proportion of specimens reproducing asexually is large, this process, which does not require synchronisation, could also explain the occurrence of juvenile and adult specimens at all times.

### 4.4. Consequences for proxies and biogeochemical cycles

As previously explained, the signals of synchronised reproduction and OVM in our data are embedded within a large overall variability of the distribution of planktic foraminifera individuals. Except for *O. universa*, we estimate that more than half of planktonic foraminifera individuals in the studied size range do not follow the canonical OVM and synchronisation trajectory. A large portion of these specimens will still reach the sediment, as evidenced by the presence of empty shells in the deepest tow intervals at all times during the cruise (Figure 3),

which will have consequences for the interpretation of the origin of the geochemical signal preserved in foraminifera shells in the sediment.

It has for example been shown that small specimens of foraminifera generally have a lower $d^{18}O$ and $d^{13}C$ value than larger specimens of the same species, among other factors also likely due to the OVM throughout their life (Berger et al., 1978; Hemleben et al., 1985). Therefore, the existence of different ontogenetic trajectories leading

to the flux of empty shells within the same population should translate into a large variability in the geochemical composition of individual shells superimposed on an overall average signal, consistent with the canonical ontogenetic trajectory. Indeed, Haarmann et al. (2011) report a large range of Mg/Ca derived calcification temperatures for single specimens of *Globigerinoides ruber ruber* obtained from the same sample, which can be explained if the specimens followed different OVM trajectories. Similar observations were made in combined

analyses of Mg/Ca and $d^{18}O$ in fossil specimens of (among others) *G. ruber ruber*, reporting unexpectedly large variability among the individual foraminifera, which appeared strongly affected by seasonality and/or the depth habitat (Groeneveld et al., 2019). Our results indicate that such variability may result from variability in ontogenetic trajectories among individuals within the same population. The same phenomenon would also affect variability in the geochemical composition of successive chambers in the same individual. The canonical trend

reflecting the optimum OVM trajectory would emerge on the background of variability in OVM trajectories recorded among the individual specimens. In both cases, the reproductive strategy model revealed by our study





implies that geochemical analyses made on a large pool of specimens should reveal a signal of average calcification depth and time consistent with the canonical ontogenetic trajectory, whereas analyses of single individuals should reveal a large variability, even when the collected specimens originated from the same population.

In addition to the consequences for the interpretation of paleoceanographic proxies, the existence of synchronised reproduction should induce short-term dynamics in pelagic carbonate flux, for which planktonic foraminifera are one of the major contributors (Schiebel, 2002). Indeed, in the equatorial ocean, synchronicity (lunar or other) in the reproduction of e.g. *G. ruber* and *O. universa* has been invoked as an explanation for a periodicity in the flux of empty shells intercepted in sediment traps (Kawahata et al., 2002; Venancio et al., 2016). This suggests that the already well documented seasonal variability in species flux (e.g., Mohiuddin et al., 2004; Lončarić et al., 2005; Salmon et al., 2015) is likely composed of regular short pulses of disproportionately large amounts of disproportionately large specimens. The consequences of such a pulsed pattern of pelagic carbonate export flux for biogeochemical cycling such as the biological carbonate pump remain an important target for future studies.

**Conclusions**

Using new direct observations, the first from a tropical open-ocean setting, we were able to test for the existence of a synchronised reproduction and ontogenetic vertical migration in four symbionts-bearing species of planktonic foraminifera representing all three main clades.

Our observations of abundances, presence/absence of cytoplasm and size measurement for the species *Globigerinita glutinata*, *Globigerinoides ruber ruber*, *Globorotalia menardii* and *Orbulina universa* revealed a constant dominance of small specimens and the presence of living and dead specimens within all size fractions at all depths and time. However, superimposed on this pattern was a subtle but statistically significant signal indicating disproportionality in the abundance of individuals within specific size fractions with depth and through time in a way that is consistent both with synchronised reproduction and ontogenetic vertical migration via descent of maturing individuals (Figure 7).

Our data show an overrepresentation of large specimens of *G. ruber ruber* below 20 m and of *G. glutinata*, *G. menardii* and *O. universa* below 40 m depth at the beginning of the cruise (shortly after full moon) and an overrepresentation of small specimens in the surface layer above these depths (Figure 7) five to eight days later. These observations imply that planktonic foraminifera population dynamics in the open ocean (at least for the studied species) involves synchronised reproduction and vertical migration of adult specimens. Since our sampling covered a substantial area, this pattern must affect planktonic populations regionally in the same way.

In all four species, we observe descent of mature specimens during ontogeny (OVM), which we attribute to a combination of passive processes that allow each planktonic foraminifera species to reach and sustain a specific depth of interest (preferential depth, in our study not tied to the MLD nor to the DCM) where they undergo gametogenesis. We speculate that descent (rather than ascent) of mature specimens is the "common" direction of OVM in the open ocean, at least for species bearing symbionts, and that motility of gametes may aid in depth restitution of juvenile specimens.

As we show that a large fraction of the population does not follow the canonical trajectory, sedimentary assemblages probably contain a mixture of specimens recording a range of individual trajectories in time and space, among which the canonical trajectory emerges only as a slight but significant disproportionality in





abundances. This is because next to the canonical trajectory, other reproduction triggering parameters and reproduction types (e.g. asexual reproduction) may occur.

The model extracted from our dataset with the canonical signal of synchronised reproduction and OVM occurring on the backdrop of other ontogenetic trajectories (Figure 7) helps reconcile previous contrasting observations. It implies that in the sedimentary record, fossil shells of foraminifera carry a signal of different vertical life trajectories, and that the background signal of synchronised reproduction that emerged may also generate a pulse in the periodicity of the flux of mature specimens to the deep ocean and thus impact the short-term carbonate export.


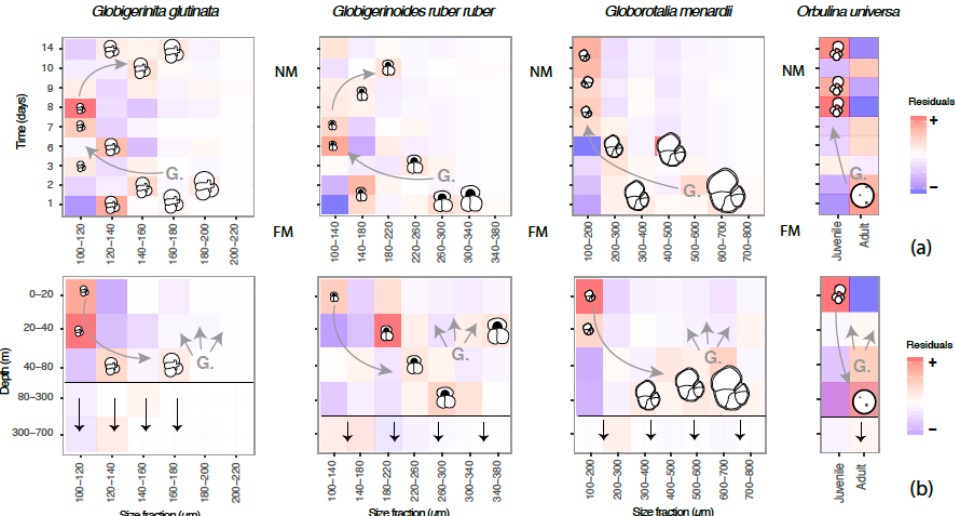

Figure 7: A model describing the canonical ontogenetic trajectory of *G. glutinata*, *G. ruber ruber*, *G. menardii* and *O. universa* as recorded in the studied material. The residuals (further discussed in 2.d and 3.4) highlight size classes that are over-represented (red tones, "+") or under-represented (blue tones, "-") across time (a), FM=Full
Moon and NM=New Moon) and depth (b) in comparison to the whole dataset. Every coloured square = presence of specimens and show that foraminifera of all size were collected at almost all depth and time throughout the survey. Based on our observations the horizontal black lines in b. represent the depth from which >50% of specimens are dead and the vertical black arrows illustrate the flux of dead shells. Based on our interpretations (see results and discussion) "G." in both panels hypothesize gametogenesis and the presence of gametes followed
by gamete dispersions (associated grey arrows). The curved grey arrows highlight the hypothetical ontogenetic trajectories of specimens (gametes to reproductive stage) across time and depth.

**Appendices**

Table A1: Size variability (minor axis, in μm) of the measured specimens of planktonic foraminifera per species
over the study when all and only specimens larger than 100 μm are considered.

| | | *Globigerinita glutinata* | | *Globigerinoides ruber ruber* | | *Globorotalia menardii* | |
|---|---|---|---|---|---|---|---|
| | | All | > 100 μm | All | > 100 μm | All | > 100 μm |
| | n | 3520 | 2734 | 1073 | 762 | 1085 | 746 |



| | | | | | | |
|---|---|---|---|---|---|---|
| | Minimum | 37.05 | 100 | 20.65 | 100.1 | 59.35 | 100.1 |
| Minor axis (µm) | Median | 113.99 | 118.7 | 114.38 | 122.3 | 116.19 | 134.9 |
| | Mean | 114.36 | 122.6 | 119.21 | 134.6 | 160.71 | 195.4 |
| | Maximum | 216.1 | 216.1 | 388.53 | 388.5 | 790.89 | 790.89 |

Table A2: Number of living specimens (N) and mean size (minor axis, in µm) of planktonic foraminifera per species and depth environments.

| | | *Globigerinita glutinata* | | *Globigerinoides ruber ruber* | | *Globorotalia menardii* | |
|---|---|---|---|---|---|---|---|
| Environment | Depth interval (m) | N | mean | N | mean | N | mean |
| Surface | 0-20 | 995 | 107.4 | 357 | 98.3 | 805 | 135.1 |
| Supra epipelagic | 20-40 | 525 | 107.8 | 109 | 111.3 | 78 | 180.1 |
| Epipelagic | 40-80 | 209 | 102.2 | 51 | 125.7 | 51 | 293.6 |
| Infra epipelagic | 80-300 | 203 | 114.9 | 132 | 133.4 | 64 | 305.6 |
| Mesopelagic | 300-700 | 49 | 114.5 | 18 | 135.0 | 29 | 186.4 |

Table A3: Number of living specimens (N) and mean size (minor axis, in µm) of planktonic foraminifera per species and station.

| | *Globigerinita glutinata* | | *Globigerinoides ruber ruber* | | *Globorotalia menardii* | |
|---|---|---|---|---|---|
| Station n° | N | mean | N | mean | N | mean |
| 2 | 120 | 116.4 | 44 | 149.9 | 110 | 426.3 |
| 3 | 134 | 109.9 | 68 | 120.9 | 20 | 361.8 |
| 4 | 79 | 113.6 | 24 | 124.7 | 18 | 271.7 |
| 6 | 33 | 94.1 | 4 | 119.0 | 3 | 335.3 |
| 7 | 163 | 103.1 | 30 | 108.0 | 3 | 191.0 |
| 8 | 295 | 99.4 | 110 | 109.6 | 37 | 126.7 |
| 9 | 134 | 100.1 | 81 | 95.8 | 48 | 107.0 |
| 10 | 764 | 112.4 | 182 | 108.9 | 567 | 110.7 |
| 12 | 259 | 106.3 | 124 | 101.1 | 221 | 133.6 |

Table A4: Results of the Kruskal-Wallis tests with Dunn Sidak correction (DS) for the three species of studied planktonic foraminifera with the comparison groups A and B for all stations (2, 3, 4, 6, 7, 8, 9, 10, 12). Statistically
significant results are highlighted in grey and with bold characters.

| | | *p*-value (DS correction) | | |
|---|---|---|---|---|
| group A | group B | *Globigerinita glutinata* | *Globigerinoides ruber ruber* | *Globorotalia menardii* |
| 2 | 3 | 0.608080093 | **0.035932605** | 0.999999479 |
| 2 | 4 | 1 | 0.723194198 | 0.999999485 |
| 2 | 6 | **3.36203E-05** | 0.982503452 | 0.99437856 |
| 2 | 7 | **0.000144447** | **2.75833E-06** | 1 |
| 2 | 8 | **1.70995E-10** | **0.000221489** | **0** |
| 2 | 9 | **1.02211E-07** | **9.74181E-11** | **0** |
| 2 | 10 | 0.996222105 | **3.02309E-09** | **0** |





| group A | group B | Globigerinita glutinata | Globigerinoides ruber ruber | Globorotalia menardii |
|---|---|---|---|---|
| 2 | 12 | **0.003908006** | **3.43097E-09** | **0** |
| 3 | 4 | 0.989337609 | 1 | 1 |
| 3 | 6 | **0.015508053** | 1 | 0.999997473 |
| 3 | 7 | 0.493132702 | 0.123828828 | 1 |
| 3 | 8 | **0.000252471** | 0.999987102 | **1.24834E-05** |
| 3 | 9 | **0.004864475** | **0.001511193** | **1.54061E-10** |
| 3 | 10 | 0.996326771 | **0.048019673** | **6.84652E-12** |
| 3 | 12 | 0.998621508 | **0.032419241** | **0.012379879** |
| 4 | 6 | **0.00045299** | 1 | 0.999998622 |
| 4 | 7 | **0.010736135** | 0.225192202 | 1 |
| 4 | 8 | **1.88625E-06** | 0.999910131 | **3.73722E-05** |
| 4 | 9 | **4.49042E-05** | **0.030466022** | **1.1917E-09** |
| 4 | 10 | 1 | 0.315264384 | **1.36179E-10** |
| 4 | 12 | 0.139075664 | 0.224002248 | **0.027200628** |
| 6 | 7 | 0.688875924 | 0.999999997 | 1 |
| 6 | 8 | 0.999948837 | 1 | 0.999634161 |
| 6 | 9 | 0.999983077 | 0.99999991 | 0.795884885 |
| 6 | 10 | **0.000141565** | 1 | 0.937995383 |
| 6 | 12 | 0.124403244 | 1 | 1 |
| 7 | 8 | 0.823034275 | 0.577023078 | 0.746718316 |
| 7 | 9 | 0.977102329 | 1 | 0.158783557 |
| 7 | 10 | **6.85565E-05** | 0.99999987 | 0.282176207 |
| 7 | 12 | 0.999719084 | 1 | 0.99996622 |
| 8 | 9 | 1 | **0.021056195** | 0.840225369 |
| 8 | 10 | **0** | 0.490109341 | 0.992377928 |
| 8 | 12 | 0.005837358 | 0.348936346 | **0.0395251** |
| 9 | 10 | **4.3455E-09** | 0.978892147 | 0.999977736 |
| 9 | 12 | 0.092684766 | 0.999895592 | **8.74673E-09** |
| 10 | 12 | **0.002853583** | 1 | **0** |

Table A5: Results of the Kruskal-Wallis tests with Dunn Sidak correction (DS) for the three species of studied planktonic foraminifera, with the comparison groups A and B for the depth comparison (1 = surface, 2 = supra epipelagic, 3 = epipelagic, 4 = infra epipelagic, 5 = mesopelagic, as defined in section 2.3 and in Table A2). Statistically significant results are highlighted in grey and with bold characters.

| | | $p$-value (DS correction) | | |
|---|---|---|---|---|
| group A | group B | Globigerinita glutinata | Globigerinoides ruber ruber | Globorotalia menardii |
| 1 | 2 | 0.999996701 | 0.088352274 | 0.977065781 |
| 1 | 3 | **0.01720279** | **1.19095E-06** | **0.000614052** |
| 1 | 4 | **1.05374E-05** | **0** | **1.00345E-08** |
| 1 | 5 | 0.224765181 | **1.04835E-06** | 0.000757423 |
| 2 | 3 | **0.016029819** | **0.027355658** | 0.102431327 |
| 2 | 4 | **0.000161843** | **5.92648E-11** | **0.000641916** |
| 2 | 5 | 0.329156978 | **0.000851265** | **0.037785271** |
| 3 | 4 | **4.55959E-09** | 0.184506327 | 0.945601843 |
| 3 | 5 | **0.003616986** | 0.529742148 | 0.998132028 |





| 4 | 5 | 0.999999412 | 0.999981711 | 0.999999989 |

**Data availability**

Planktonic foraminifera counts, species concentrations and size will be made available on request to the main author until their online publication on PANGAEA (www.pangaea.de).


**Author contribution**

M. Kucera and M. Siccha conceptualised the data collection. J. Meilland and M. Kaffenberger processed the samples and performed foraminifera shell measurements. J. Meilland did the planktonic foraminifera taxonomy M. Siccha performed the statistical analyses. J. Bijma provided support for the data processing (residuals and

mortality) and interpretation. The manuscript was written by J. Meilland with the help of all co-authors who approved its final version.

**Competing interests**

The authors declare that they have no conflict of interest.


**Acknowledgements**

The authors thank the captain, crew and participants of the RV Meteor expedition M140 FORAMFLUX for their assistance with plankton sampling and CTD measurements. We acknowledge the technical help provided by Sylvia Malagoli in processing some of the samples and helpful discussions with Lukas Jonkers for data

visualisation. We thank the Deutsche Forschungsgemeinschaft (DFG; Geschäftsstelle Deutsche Forschungsschiffen) for financing the research project of Julie Meilland ("FoPa", project ME 5192/1-1) and the Cluster of Excellence "The Ocean Floor – Earth's Uncharted Interface" who's financial support in our laboratories infrastructures contributed to the success of this research.

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
