# Peer review of "Population dynamics and reproduction strategies of planktonic Foraminifera in the open ocean"

_Biogeosciences, 2021_

## Author Comment (AC1)

Dear Ralf,

I would like to thank you for the time you devoted to our manuscript and for the very constructive comments and corrections you provided. Please find hereby a point-by-point response to your review, in blue characters.

As you will see, we took most of your remarks in consideration and amended the manuscript accordingly.

We hope this latest version will satisfy you.

Best wishes,

Julie, on behalf of all co-authors.

N.B: the line numbers provided in our response corresponds to the line numbers of the edited manuscript (track change).

Referee #2:
The manuscript of Julie Meilland and coauthors on "Population dynamics and reproduction strategies of planktonic Foraminifera in the open ocean" adds valuable perspectives to the discussion on the reproduction strategy of planktic foraminifers. Assemblage data from plankton tows sample in-situ populations are suited for analyses of the respective population dynamics. Using the same statistical approaches, which have been applied in earlier studies facilitates comparability, and confirm earlier results and interpretations.
Despite the 14-day long data set that "only" covers half of a synodic lunar cycle, i.e., half of the full reproduction cycle assumed to be executed by most modern planktic foraminifers, the paper presents valuable assumptions on the reproduction systematics of four species.

We know the referee has himself worked extensively on the topic during his career and therefore thank him for this positive and encouraging statement.

Overall, the paper is well written concerning the reproduction strategies of modern planktic foraminifers. In contrast, the section 4.4. on "Consequences for proxies and biogeochemical cycles", which discusses the paleoceanographic implications of the finding presented here, is rather poorly executed, and may be rewritten, or removed from the manuscript. I would suggest publication of the paper following some larger and smaller improvements given in the following.

We understand the referee's remark about section 4.4. This section is important to us, we therefore decided to keep it but to rephrase it partly (L.707 to L.751). We also toned down the title by writing "implications" instead of "consequences".

1. More general comments
Plankton are present in the ocean at all times. As ecological conditions change, small (juvenile, dormant, or resting) stages of some species may profit and start growing to be eventually

sampled by plankton nets of a certain mesh size. The required ecological conditions do not possibly occur at predefined water depths intervals, but may vary across the regional and global ocean. This is possibly also the case in the study presented here. Whereas it is implied in the paper (although not explicitly stated) that the region sampled and analyzed here may be characterized by rather homogenous hydrologic conditions, this may not be true for the ecological conditions affecting the population dynamics and (assumed) systematic reproduction of planktic foraminifers. As can be seen for the Figure 2 (panel A) of the manuscript, the region is heavily affected by eddies (Fig. 1). Both cold and warm core eddies are characterized by upwelling and downwelling at their centers and margins, which affects trophic state and vertical transport of waters bodies and the plankton included within. Vertical transport of planktic foraminifers both up and down the surface water column may consequently act here in the same way as it does in other regional of the ocean, and would affect the population dynamics and interpreted reproduction scenarios discussed here.

We agree with the referee, the region is highly dynamic and we emphasize potential vertical and lateral transport more in section 2.1. (L.149 to 152 and L.174) and in section 4.1. (L.497 to 501), stressing that our observations reflect the dynamics of multiple populations rather than of a single one.

Ontogenetic terms such as juvenile and adult are not used in the correct way in many places of the manuscript, and may need to be changed to "small" or "large" (e.g., lines 109, 226, 281). Just assuming that individuals of a certain size of a certain species would be "adult" in the sense of being capable of reproduction, as in the Figure 6, is speculation.

We agree and respectively replaced "juvenile" and "adult" by "small" and "large" when it was necessary, as suggested by the referee.

Proof of reproduction may only come from the presence (or absence) of gametogenetic (GAM) calcite on top of the shell (e.g., G. bulloides, Schiebel et al., 1997). Unfortunately, GAM calcification does not occur in some species such as G. ruber, and size of earliest possible reproduction may be identified from population dynamics. There is no proof for the assumption that reproductive maturity is reached as late as in the very large size classes as shown in Figure 6. If this would be the case, reproduction would be possible only in the few specimens that grow very large. Alternatively, the change from "mortality" to "reproduction" may occur at the size class from 140-180 microns and 200-300 microns in G. ruber and G. menardii, at which size the adult stage is reached.

We thank the referee for drawing our attention to this fair point. It is indeed a difficult "cut" to make without having a clear sign of reproduction as the GAM calcite mentioned by the reviewer. We therefore rephrased a bit saying that the value given in Figure 6b are "conservative" or "safe" but precising that one cannot exclude that reproduction might already have start in the $140 - 180$ µm for *G. ruber ruber* and in the $300 - 400$ µm for *G. menardii*.

In O. universa, "trochospiral and spherical" may be the correct term (line 209). Please see also lines 411-412, and 415, where speculation about the unproven connotation of test size fraction and ontogenetic stage is repeated.

We replaced the terms "juvenile" and "terminal" by "trochospiral" and "spherical" throughout the manuscript, as advised by the referee and speculations about test size fraction and ontogenetic stage have been toned down throughout the manuscript based on the referee's comments.

2. More specific comments
Line 168: "concentrations" may be changed to "standing stocks" in case of live assemblages

Changed

Line 311: the statistical significance should be proven by numbers, and not assumed

We agree with the referee and statistical significance is based on numbers throughout the entire manuscript (Table A4 and A5).

Line 387: "reproductive mortality" this is a strange term for the process you want to describe here. Please google "reproductive mortality", which has a completely different connotation. I would suggest to simply use "reproduction" instead.

The term has been changed.

Lines 395-396: This may be written in passive, since it may not be the foraminifer's decision: "This is because the life of a foraminifer ends at gamete release."

We used the referee's suggestion.

Lines 425-426: Better start sentence with: "Small individuals..." The size class >100 microns does possibly not include juvenile stage of most of your species, but rather neanic and adult stage only. Please have a look at Schiebel and Hemleben (2017, and references therein); most importantly the papers of Geert-Jan Brummer.

We used the referee's suggestion and now start the sentence by "Small individuals…"

Lines 428-429: „The constant presence of juveniles and dead specimens of foraminifera from all species suggest that reproduction may have occurred continuously during our survey."
This is a misconception. This only shows that plankton grows and dies at all times.

We understand the referee's suggestion however the sentence is very speculative.

Line 440, and other places: The concept of synchronized reproduction was possibly introduced by Ahuve Almogi-Labin (1984); see also Erez et al. 1991; the first ideas on this may have emerged as early as in 1967 from Berger and Soutar...

We replaced "introduced" by "support".

Line 445: We have learned from Spero et al. (2015) that the spherical chamber of O. universa

may include up to seven day and night layers of calcite, which means that gametogenesis may not really be imminent upon first formation of the spherical chamber.

We added the reference and replaced "imminent" by "relatively close".

Lines 477-478: „This contrasts with the OVM pattern suggested for G. menardii by Schiebel and Hemleben (2017),…" These patterns vary with ecological conditions. I have found more individuals of G. menardii at greater depth, which would also largely exclude photosymbiont activity.

We added a sentence saying that the discrepancy between your observations and ours for *G. menardii* could suggest that the OVM pattern varies regionally and based on the ecological conditions.

Lines 482-482: According to Takagi et al. (2019), photosymbiosis in G. menardii is merely facultative.

We specified that photosymbiosis in *G. menardii* is facultative.

Line 496: This is possibly the latest (and only secondary) reference of many earlier (and original) references.

Indeed, this and earlier studies are presented in the introduction as they are the ones who "set the scene" but we deliberately choose in the discussion to focus more on the studies of Bijma, Hemleben and yours as it allows for more direct comparisons. Your study also has the advantage to not be limited to the Red Sea where most of the trajectory of OVM coupled to lunar synchronicity has been studied and discussed. We also indeed tend to cite Schiebel and Hemleben 2017 extensively instead of all the "original references" as the book does a great job at providing the most complete "update" on this ecological question and as one could refer it.

Lines 529-535: Why should gametes develop this strange behaviour and escape from the place where they are released? Gametes are possibly released at certain depths to provide them with optimum conditions for survival, and straight ascent would decrease the survival rates. This makes no sense.

We understand the referee's point and we shortened and rephrased the section from L.638 to L.641.

Line 549: Survival of a population would be ensured if only one offspring of one parent would make it to reproduction, which would be much less than 5 % in case of 100,000 offspring.

This is correct in the case of asexual reproduction (only weakly documented thus far) but not for sexual reproduction that would need gametes release from two specimens in order to produce at least one successful zygote (extensively discussed in Weinkauf et al., 2020, in discussion).

Lines 571-572: „… but often interpreted literally, assuming that all specimens follow the depicted ontogenetic trajectory." This is possibly your very personal interpretation of the literature.

We rephrased the sentence.

Line 619: Why ALL 3 clades? How many clades are there according to your information? I would count on 4, which includes the Hastigerinidae.

We agree with the referee, this is why we systematically specified "main" clades as the fourth one is only constituted of the Hastigerinidae. To avoid some misunderstanding we specified it.

Figure 7: I read the depth trajectories in the opposite direction. Reproduction may occur around day 1 or 2 near the thermocline / DCM. From day 6 or 7 (3 at the earliest in G. glutinata), more small individuals occur in the overlying water column; these small individuals, however, did already grow to the size of >100 microns. In the following, increasingly more larger individuals of G. ruber and G. glutinata occur in the surface water column. This is quite similar to the development of G. bulloides in the NE Atlantic (Schiebel et al. 1997). Larger G. menardii did not occur in the surface water column; this is quite similar to what I have seen in G. menardii (Schiebel and Hemleben 2017).

The complete reproductive cycle may indeed as well be a complete physical cycle in the water column. We hereby present one half of it and the referee's suggestion illustrates the second half. As we however do not have reliable data below 100 μm we cannot trace the trajectory of these small individuals and would prefer to only show the "descending" part of the cycle here.

---

## Author Comment (AC2)

Dear Referee #1,

We warmly thank you for the constructive and helpful comments you provided in order to improve our manuscript. We took most of your remarks into consideration and hope this latest version of the manuscript will satisfy you.

Each of your comment has been answered in blue characters in the following document.

Kind regards,

Julie Meilland, on behalf of all co-authors.

N.B: the line numbers provided in our response corresponds to the line numbers of the edited manuscript (track change).

General comments:

The manuscript entitled "Population dynamics and reproduction strategies of planktonic foraminifera in the open ocean" by Meilland et al. examined the presence, pattern and extent of synchronised reproduction and ontogenetic vertical migration of planktonic foraminifera, the phenomena which have long been discussed since the earliest study of this taxon and always controversial with evidence both in favor and against on. Their finding suggested the presence of synchronised reproduction and ontogenetic vertical migration, superimposed on the large fraction of the population that does not follow the canonical trajectory. The manuscript is well-written, and carefully discussed with adequate data analysis and statistics. This study has fundamental importance not only to help us understand the population dynamics of planktonic foraminifera but also their sedimentary assemblages; what is recorded and how to extract the canonical trajectory from fossil samples.

It was my great pleasure to review this manuscript. I recommend publication after the authors address the issues I have outlined below.

We thank the referee for this positive evaluation of our research.

 Major points:

1. Size measurement protocols

It would be helpful to have a representative series of images showing the size measurement (image processing) procedure, maybe in the supplement. Is it possible to automatically extract shell outline even for specimens with densely radiated spines? Does the "minimum diameter" mean minimum Feret diameter?

To allow the reader understand our segmentation procedure, we now provide images of it using the example of *G. glutinata* in Figure S1. We also added a sentence in the section 2.2. to specify what the "minimum diameter" corresponds to (L.213 to 215). More details are also provided in our reply to the referee's second comment in the "Minor points". Among the species analysed, only *G. ruber ruber* has spines and the vast majority of them were broken

during the sampling and throughout samples preparation (picking of the specimens and positioning on customed slides for the size measurements). The shell outline provided for *G. ruber ruber* therefore does not include the spines.

2.  Effective digit

What is the error range of the size measurement and the effective digit? In Table A1, some are shown with two decimal places (e.g., 113.99, 790.89). Please align the number of digit after the decimal point based on the effective digit.

The absolute accuracy is of 0.6 μm that one can round up to an error range for the size measurement of 1 μm for a single measurement (i.e. per specimen). We added this information in section 2.2., L. 214 to 215. We therefore agree that the digit in Table A1 should not remain and we removed it where needed.

3.  Size class intervals

I think the size class intervals used here are fine, but how did you determine the interval (or the number of category). Here the size of G. glutinata alone is divided into 6 (but in Figure 6 the largest class omitted), whereas the others are 7.

We decided to choose size classes that encompassed the same relative range of distribution namely from 100 μm (minimum mesh diameter) to the largest specimen found without increasing the number of classes significantly per species and without introducing empty classes. In Figure 6, the largest size class of *G. glutinata* is not represented as the relative mean abundance of specimens observed in 200-200 μm is very low (<0.5%) and would not allow for a correct estimation of the relative mortality.

4.  Calculation of abundances

Did you used a flow meter for the calculation of towed water volume or just used the net aperture area and towed depth? Please specify. If the latter, it is calculated on the assumption that the extent of net clogging is similar among nets.

The multinet was equipped with a flow meter and allowed us to directly determine the volume of filtered water for each net (L171 to 172).

5.  The data under 100um

It is rather surprising that the estimated minimum size of maturity in G. glutinata is smaller than 100um. As is written in the text, a large proportion of specimens is smaller than 100um and hence excluded from the analysis for calculation of residuals and mortality. I understand why the authors hesitate to use the smaller size classes since the net mesh was 100um. Although, as I wrote above, if the towed water volume is not calibrated using a flow meter, the net clogging is regarded as the same in this data analysis in the first place. In any case, it is worthwhile to show, in the supplement, the data smaller than 100um and include in the mortality figure and residual figure. I would recommend including it, at least for G. glutinata.

We appreciate the Referee's suggestion but without having a precise idea of the representativeness of the fraction of individuals caught by the net below 100 µm size, we would feel uncomfortable presenting these data in details.

6. Background population that does not follow the canonical trajectory

One of the importance of this paper is that they clearly showed that a large population does not follow the canonical trajectory. Then, do you think the background population succeeds in reproduction without synchronizing time and space, or they are just the "leak" of canonical population and less likely to succeed in reproduction (such as abortive migration in fish)? You mention in the abstract that "reproduction might have occurred continuously", so the former would be your idea, I suppose. Then how? Does it contradict the Weinkauf et al. (2020) emphasizing that spatial and temporal synchronization is inevitable for maintaining of the population?

We indeed think that the background population could succeed in reproduction without synchronizing on a large scale but within patches (more information about patchiness in Siccha et al., 2012 and Meilland et al., 2019) and/or using asexual reproduction. The first hypothesis could be the result of an event or situation that would trigger gametogenesis locally. We do not think this contradict the work and hypothesis formulated in Weinkauf et al., 2020 (still in discussion) but rather that the "truth" sits in between and that population dynamics in planktonic foraminifera most likely is the result of various triggers and reproduction modes.
* * *
Minor points:

Line 96: Takagi et al. 2020 ---> Maybe Takagi et al. 2019?

Indeed, we corrected the citation L.97.

Line 203: the minimum diameter ---> Did you used the Feret diameter? Please specify because there are many ways to measure diameter.

The minimum diameter here refers to the "minor axis length" which corresponds to the length (in pixels) of the minor axis of the ellipse that has the same normalized second central moments as the region, returned as a scalar. We now specify it L.213.

Line 284: 114.5 ---> Referring the Table A1, the original number is 114.38. Since the others are rounded to integer, it should be 114 here.

Corrected

Line 345: method section (2 d) ---> ? (the same "2.d" is in the caption of Figure 7)

We replaced "(2 d)" by the section it referred to in the methods (2.3.).

Line 392: Because of the overproduction of gametes per individual, the mortality in planktonic foraminifera is expected to be very high among the smallest size class ---> It should be so. But it sounds that gametes are the initial population of planktonic foraminifera which is not true (zygotes are the initial smallest class of population). How about saying like "Because the zygotes (youngest individuals) are overproduced per individual even with the limited rate of reproductive success (a mean of 21 zygotes per individual in the entire population, Weinkauf et al., 2020), the mortality in planktonic foraminifera is expected to be very high among the smallest size class".

We agree and we rephrased using the referee's sentence suggested.

Line 409: studies size range ---> studied size range

Done

Line 423: Tâ    and Salinity ---> temperature and salinity

Done

Line 459: manypopulations ---> add a space

Done

Line 481: this OVM pattern ---> the ascending OVM pattern

Done

Line 503: This theory corroborates ….. et al., 2021) ---> Are there any papers of this kind for warm water species? Since these studies are on cold water, non-symbiotic species, and more directed on the ocean acidification topics, it would be better to cite something else.

We know about the paper of e.g. Marshall et al., 2013 on *T. sacculifer* and *G. ruber* however, we still think that the citations we used, even if on *G. bulloides* and *N. pachyderma*, are the best suited ones to discuss increasing shell density with size and depth the way we intend to.

Line 516: the properties of fibrillar bodies hypothesised to help foraminifera maintain their vertical position ---> Indeed the function of the fibrillar bodies has been speculated to be linked to the function of buoyancy. However, recently, LeKieffre et al. (2020) suggested that the fibrillar bodies are the organelle for organic matter synthesis and storage prior to chamber biomineralization. So this possibility can be deleted.

We would prefer to keep this hypothesis as the paper from Le Kieffre et al., 2020 does not fully exclude a potential motility role of the fibrillar bodies.

Figure 4: Are the whiskers shown in broken lines? It is better to use normal (full) line which is easier to see.

We replaced the broken lines by full lines in Figure 4.

I hope my comments above would be helpful.

Very much so! We warmly thank you for the time devoted to our manuscript.